# Development of Education Field Student Digital Competences—Student and Stakeholders' Perspective

**Edīte Sarva \*, Gatis Lāma, Alise Oļesika, Linda Daniela**  **and Zanda Rubene**

Faculty of Pedagogy, Psychology and Art, University of Latvia, 1586 Rīga, Latvia; gatis.lama@lu.lv (G.L.); alise.olesika@lu.lv (A.O.); linda.daniela@lu.lv (L.D.); zanda.rubene@lu.lv (Z.R.)
\* Correspondence: edite.sarva@lu.lv

**Abstract:** Alongside reading, writing, and numeracy digital literacy has become an increasingly crucial component of functional competences in the modern era. Digital competences are essential for educators who wish to stay updated with the changing needs of their students and the education sector. These competences can help them engage their students better, improve their teaching effectiveness, and advance their careers. Considering the ongoing changes in both the field of education and the field of technologies, it is important to explore the current needs for improving digital competences of educators including students in the field of education, to provide them with the necessary support. This study aims to develop recommendations for improving digital competence of educators by combining students' self-assessment through surveys with stakeholder opinions concerning digital competence development of educators ascertained in a focus group discussion. Results of the research reveal that education students do not have a statistically different self-assessment of their digital competences than students from other study fields. And although students evaluate their digital competences as high, stakeholders point out that there is a lack of digital competences to carry out technology-enhanced learning and a negative attitude towards digitalization of education among all educators' age groups. Improving the availability of digital resources and technological support as well as incorporating organisational level strategies that require application and improvement of digital competence could remedy the issue.

**Keywords:** digital competences; pedagogical-digital competence; higher education; digital skills; transversal skills

## 1. Introduction

The United Nations Educational, Scientific and Cultural Organisation (UNESCO) defines digital competences as a range of abilities to use digital devices, communication applications, and networks to access and manage information, enabling people to create and share digital content, communicate and collaborate, and solve problems for practical and creative self-fulfilment in life, learning, work, and social activities. This means that essential functional competences required to utilise digital devices and online applications effectively, are widely considered critical components of a new set of literacy competences in the digital era, along with traditional reading, writing, and numeracy competences [1]. To thrive in the connected economy and society, digital competences must also be complemented by other abilities or skills, such as strong literacy and numeracy, critical and innovative thinking, complex problem-solving, collaborative ability, and socio-emotional competences [1].

The aforementioned factors suggest that digital competences should not only be perceived as technical skills, but also focus more on cognitive, social and emotional aspects of working and living in a digital environment. Digital competences are a multifaceted moving target that is constantly evolving as new technologies emerge [2].

According to UNESCO's report "Futures of Education: A New Social Contract in Education" in 2021 global digital technologies, tools and platforms can be adapted to

support human rights, enhance human capabilities, and facilitate collective action in the directions of peace, justice, and sustainability. Beyond supporting universal access to technology, education systems are justifiably striving to develop the digital competences that are required by learners to make meaningful use of technology. There is nothing 'native' or 'natural' about these abilities. They are constructed and refined over time through intentional educational interventions alongside various forms of informal and self-directed learning [3].

Additionally, to support the sustainable and effective adaptation of education and training systems, the European Union released a Digital education action plan (2021–2027) that focuses primarily on promoting the development of a high-performing digital education ecosystem and on improving digital competences and capabilities for digital transformation [4].

Taking the aforementioned into account the digital competences of students, including those in the field of education, play an important role in this new learning paradigm. As they are the main subjects of education, students should be prepared to use digital competences responsibly and sustainably in their academic life and in their careers [5].

The aim of this study is to analyze students' self-assessment of their digital competences, to compare the results of students studying education and students in other fields, and to identify the opinion of educational stakeholders regarding the most important digital competences in order to develop recommendations for improving digital competence needed to ensure technology-enhanced learning. Therefore, the research questions are as follows:

1. What are the similarities and/or differences between the self-assessment of digital competences for students in the field of education and students in other fields?

2. What digital competences are currently seen as essential for educators by stakeholders in the field of education?

3. What are the views of professionals in the field of education on how digital competences should be developed for educators during higher education studies?

## 2. Literature Review

Digital competences are taking a key role in teacher education [6–9]. As the COVID-19 crisis has shown, the wide and sudden implementation of digital technologies highlights a number of significant risks for switching to a technology-enhanced learning environment, including differences in student skills and accessible support systems and the readiness of educational institutions and educators to shift towards a student-centred learning process, which is at the core of remote learning [10,11].

Digital competence is a potential boundary concept, as it includes several disciplines; the concept is rarely precisely defined, and representatives of different fields give it slightly different meanings [12,13]. One definition of "digital competence" states that it is a cognitive, attitudinal and technological skill that helps to alleviate many of the problems and challenges in today's knowledge society, and it has a dynamic and transversal character [14]. Another definition states its importance as soft skill in private, professional and academic life as it includes knowledge, abilities, awareness, and attitudes needed for the conscious, safe, critical and effective use of digital tools for different tasks such as problem-solving, communication, information in different environments [15,16] to contribute independently and in partnership with others in an creative, critical, responsible and responsive way [17].

In the context of education, digital competence is considered as the ability, along with a strong theoretical foundation, investigation and experimentation to apply the knowledge, attitudes and skills necessary to plan, implement, evaluate and continually review ICT-supported teaching and learning processes [18] and the ability to understand and express the transformation of information into knowledge, operations, and services by making analytical, productive, and creative use of ICTs and social software [19–21] and it significantly affect their academic achievement [22]. The concept of pedagogical-digital competence has been proposed to describe the knowledge, skills, and attitudes that teach-

ers need to effectively integrate digital technologies into their teaching practices [18]. It includes understanding how to use digital tools to support the achievement of learning objectives, creating and adapting digital learning resources, selecting appropriate technologies for specific teaching contexts, and fostering digital literacy and digital citizenship in students. Pedagogical-digital competence goes beyond simply knowing how to use technology—it involves understanding how to use technology in a way that enhances the teaching and learning process. This includes understanding how to use technology to differentiate instruction for diverse learners, to provide opportunities for collaboration and communication, and to support student-centred learning [23].

Additionally, digital competence does not need to be perceived as a single actor but rather can be seen as a school-level characteristic (meaning that schools and universities can be digitally competent). Moreover, schools and universities, including strategic leadership, need to become digitally competent in their ways of structuring and organising resources and institutional infrastructures to enable their staff to do the same [12,24–27].

Research shows that educators feel safer using and are more interested in DS (digital solutions) that replicate existing learning experiences than those with which they are not yet familiar or those which strongly change teaching and learning dynamics [28,29]. In a digital medium this means stripping much of the possible interactivity offered by DS, making learning more frontal and less engaging, which is concerning, considering how important the engagement of students is, perhaps especially in digital learning [30–32].

In this study, the researchers utilise the classification of competencies based on the "Digital Competence Framework for Citizens" [33]. Therefore, the areas of digital competence (information literacy, communication and collaboration, digital content creation, safety, problem solving), levels and behavioural indicators of this framework were adapted for this research. For students in the field of education this could also be considered as a measurement of their pedagogical-digital competences.

## 3. Methodology

The research was conducted in two stages:

(1) In the first phase, the digital competences of master and doctoral students in various fields were assessed quantitatively;

In this quantitative study, data were collected using an assessment tool for students' transversal competences developed in the ESF project 8.3.6.2: "Development and Implementation of the Education Quality Monitoring System" 8.3.6.2/17/I/001 [34,35] which was prepared in a digital form using the Question Pro platform. The tool was constructed to measure 6 different transversal competences. The principle of availability was used for the sample and a total of 686 respondents participated in the study. Questionnaire was offered during lectures as an alternative to another study task and the questionnaire was available for completion from 26 November 2020 to 13 March 2021. This article deals only with the master and doctoral students of various fields, and only includes digital competences.

For competence evaluation 7 point Likert scale was chosen as the 7 point scale provides more varieties of options which in turn increase the probability of meeting the objective reality of students [36]. Digital competences and their 5 sub-competences were evaluated using a 7 point Likert scale: information literacy (15 statements), communication and collaboration (17 statements), digital content creation (13 statements), safety (17 statements), problem solving (11 statements). The study analysed the level of digital competences of 161 master and 43 doctoral students. Students represented 5 Latvian higher education institutions (Rezekne Academy of Technology, University of Latvia, Daugavpils University, Riga Technical University and Rīga Stradiņš University). One of the questions of the research focused on assessing the digital competences of education sciences students and to comparing them with the digital competences of students from other study fields. Overall, 80 students from the educational sciences and 124 students from other fields of study participated in the survey. The participants filled out a questionnaire as part of a study module in different study programs. The questionnaire was proposed to students as an

alternative to other learning activities. Participants were selected on the basis of accessibility. The average age of the participants was 32 years (Me = 31, SD = 8.29). To determine the reliability of the Likert scale, Cronbach's alpha values were calculated separately for each of the digital sub-competences. An exploratory factor analysis was chosen to examine how the questionnaire functioned among master's and doctoral degree students. To determine whether there were significant differences between students in the educational sciences and students from other fields of study in their self- assessed digital sub-competence ranks, a Mann Whitney U test was carried out. Data were analysed using SPSS statistics version 21 and Microsoft Excel.

(2) In the second phase a focus group discussion was organised to ascertain the views on developing digital competences for future educators.

To represent the views of the different parties involved, the discussion was held amongst a doctoral student in the field of educational science, who was also an in-service teacher, a professor, a lecturer who also served as the dean of the Faculty of Pedagogy and a school principal who was also a practising teacher. To anonymize the data, the participants were coded—student (S), school principal (P), lecturer (L)—these codes are used in this article as well. The selection of participants for the focus group was made so that each level of the sides involved was represented, and convenience sampling was employed. The focus group discussion was led by one of the authors of this article.

At the beginning of the discussion a short description of digital competences was given to establish a common understanding of the topic. The moderator of the discussion then asked previously selected questions about the necessity of digital competences for teachers and its development during studies to each participant according to their area of expertise. Participants answered these questions, expressing their opinions in turns. Following this all participants engaged in a moderated conversation, exchanging their opinions and reacting to the information provided by other participants. The conversation was transcribed and summarised for further analysis. Theses were formed, summarising the views of the participants in the discussion. Discussion took place during an online meeting and was 60 min long. The conversation was held in Latvian, and for the purposes of this article, the information was translated into English by the authors of this article, attempting to preserve the opinions expressed by interviewees. The study considered all ethical research standards in accordance with the General Data Protection Regulation (GDPR).

## 4. Results

*Digital Competences Assessment*

To determine the internal consistency of the Likert scale Cronbach's alpha values were calculated separately for each of the digital sub-competences. The internal consistency of the Likert scale was found to be excellent for digital sub-competences such as communication and collaboration, safety, problem solving, while it was good for sub-competences like Information literacy and digital content creation (Table 1).

**Table 1.** Cronbach's alpha values for each digital sub-competence.

| Sub-Competence | Cronbach's Alpha | Items |
|---|---|---|
| Information literacy | 0.896 | 15 |
| Communication and collaboration | 0.931 | 17 |
| Digital content creation | 0.897 | 13 |
| Safety | 0.921 | 17 |
| Problemsolving | 0.924 | 11 |

An exploratory factor analysis was chosen to examine how the questionnaire functions among Latvian master or doctoral degree students to determine the number of factors that could be identified in the data. The KMO (Kaiser-Meyer-Olkin measure of sampling adequacy) value (0.966) is >0.8; therefore, the correlation matrix is "meritorious" [37]. To reduce the number of factors, the parallel analysis engine was used [38]. The number of

factors to retain will be the number of eigenvalues (generated from the dataset) that are larger than the corresponding random eigenvalues [39]. As a result, eight factors were retained. For interpretation, the Kaiser–Varimax rotation matrix was used (Appendix A). The results indicate that the statements that measure information literacy were mostly part of the first and seventh factors, communication and collaboration were mostly part of first and second factors, digital content creation was mostly part of first and fifth factors, safety was mostly part of first, third and sixth factors and problem solving mostly consisted of first and fourth factors. Each of the sub-competences was part of the first factor, indicating that there is a connecting element throughout all digital competences. However, for each sub-competence there was at least one more unique factor that was unique.

By analysing the self-assessments of digital competences from students in the educational sciences, it can be concluded that the mean values of all digital sub-competences are relatively high and are higher than 5 (on a 7 point Likert scale) [40–42] (Table 2).

**Table 2.** Self-assessment of educational sciences students.

| Sub-Competence | Mean | Median | St. Deviation | Variance |
| --- | --- | --- | --- | --- |
| Information literacy | 5.48 | 5.67 | 0.95 | 0.90 |
| Communication and collaboration | 5.65 | 5.65 | 0.89 | 0.79 |
| Digital content creation | 5.18 | 5.23 | 1.12 | 1.25 |
| Safety | 5.09 | 5.00 | 1.03 | 1.06 |
| Problemsolving | 5.62 | 5.69 | 0.90 | 0.81 |

This suggests that students from educational sciences perceive their digital competences to be high. Three out of five digital sub-competences have been assessed relatively similarly, with similar median and mean values for problem solving and communication and collaboration self-assessments problem solving in this research was defined as the competences required for students to diagnose and address technical challenges related to the use of digital technologies, as well as to assess their digital competences, their weaknesses and potential for future development. In turn, Information Literacy in this research was defined as the competences required for students to obtain, select, evaluate, categorise, store, modify, restore, and retrieve information in various formats, using different software to perform these tasks. The high mean value of the self-assessments of students studying educational sciences indicates that master's and doctoral level students feel confident in performing the aforementioned activities. Digital communication and collaboration are of major importance as digital sub-competences in the context of remote learning. Digital communication and collaboration sub-competences allow students to create their digital identity and maintain online communication by sharing information and collaborating with others using digital technologies. These competences can be very important for future teachers in capably managing digital learning spaces.

Students of educational science self-assessed their digital content creation and safety sub-competences as less developed. Although the mean values of self-assessment by education science students for the above-mentioned digital sub-competences are considered to be relatively high, digital content creation, in particular, is an essential competence that enables the development of a modern learning process. Digital content creation was defined as the competence required to develop, integrate and refine digital content, as well as to correctly refer to the source of digital content and respect copyright. To improve the learning process for students in the educational sciences, it would be necessary to focus on the activities or tasks that promote the development of this digital sub-competence.

The mean values of self-assessments of students from other fields of study are similar to the mean values of the self-assessments of students in the educational sciences. However, in each of the digital sub-competences assessed, the self-assessment mean values of students from other fields of study are lower compared to education sciences students in each of the digital sub-competences (Table 3).

**Table 3.** Self-assessment of students from other fields of study.

| Sub-Competence | Mean | Median | St. Deviation | Variance |
|---|---|---|---|---|
| Information literacy | 5.40 | 5.53 | 0.98 | 0.96 |
| Communication and collaboration | 5.34 | 5.59 | 1.15 | 1.32 |
| Digital content creation | 4.94 | 5.04 | 1.16 | 1.35 |
| Safety | 4.86 | 4.82 | 1.12 | 1.25 |
| Problemsolving | 5.49 | 5.64 | 1.12 | 1.25 |

Similar to students from the educational sciences, students from other fields of study self-assessed their problem solving, information literacy and communication and collaboration sub-competences as the most developed among all digital sub-competences. While students from other fields of study self-assessed the aforementioned digital sub-competences on average above five (on a 7 point Likert scale), they self-assessed lower with mean values below 5 for other digital sub-competences, such as digital content creation and safety sub-competences.

A Mann Whitney U test was carried out to determine whether there is a statistically significant difference between the self-assessment rankings of students of educational sciences and those of students from other fields of study (Table 4).

**Table 4.** Comparison of self-assessment rankings: students of educational sciences and students from other fields of study (Mann Whitney U test).

| Digital Sub-Competence | Group | N | Mean Rank | Sum of Ranks | U | Z | P |
|---|---|---|---|---|---|---|---|
| Information literacy | Students from other study fields | 124 | 100.55 | 12,468 | 4718 | −0.588 | 0.556 |
| | Education sciences students | 80 | 105.53 | 8442 | | | |
| Communication and collaboration | Students from other study fields | 124 | 97.35 | 12,072 | 4322 | −1.55 | 0.121 |
| | Education sciences students | 80 | 110.48 | 8838 | | | |
| Digital content creation | Students from other study fields | 124 | 97.67 | 12,111.5 | 4361.5 | −1.454 | 0.146 |
| | Education sciences students | 80 | 109.98 | 8798.5 | | | |
| Safety | Students from other study fields | 124 | 98.48 | 12,211.5 | 4461.5 | −1.211 | 0.226 |
| | Education sciences students | 80 | 108.73 | 8698.5 | | | |
| Problemsolving | Students from other study fields | 124 | 101.11 | 12,538 | 4788 | −0.418 | 0.676 |
| | Education sciences students | 80 | 104.65 | 8372 | | | |

Although the mean values of the self-assessments for education students are higher compared to self-assessments for students from other fields for each digital sub-competence, the Mann Whitney U test results indicate that the $H_0$ hypothesis—that there is no statistically significant difference between these two student groups—can not be rejected. Therefore, it can be concluded that there is no statistically significant difference between the self-assessments of students in educational sciences and students from other fields of study for all five digital sub-competences.

## 5. Focus Group Discussion

Information from the focus group discussion was analysed by using content analysis principles and several conclusions about developing digital competences for educators were formed. These conclusions are summarised below as well as quotes from the discussion participants substantiating them.

All components of digital competences are topical for educators in their daily work, and their topicality has increased during the COVID-19 pandemic. Therefore, it is important to focus on pedagogical-digital competences, rather than just digital competences, in teacher education—including the competences of using and creating new digital solutions (DS) for education. The educators must not only be able to use DS themselves, but also be able to teach the planned content using the chosen DS to their students and to do it in an integrated way.

P: "Teachers should be able to access digital content concerning their subject, collect and store data about their students, help students learn remotely and understand and observe safety in the digital medium."

L: "Teachers have to be able to choose and also create pedagogically appropriate DS for learning. DS that direct student attention towards and help them focus on learning."

P: "Digital competences are transversal skills that are required to be taught in all age groups and standards of students. Teachers should not only be digitally competent themselves, but also be able to teach their students to use DS."

Educators understand that DS should be further integrated into the learning process, combining them with face-to-face work, but are not yet prepared for this. They are aware theoretically that the changes are necessary but at the same time, they also have a negative attitude formed by a sense of insecurity. Additionally, many available DS are not suitable for education. Therefore, it is important to evaluate, choose, learn and use the most convenient DS for educators and students—the ones that facilitate everyday work, rather than making it more complicated. It is risky, if the educator has created DS that do not meet educational goals, as it can serve as further confirmation and justification for the negative attitude towards the digitalization of education.

P: "It is clear that we are not going back to how it was before the COVID-19 pandemic. Learning in the digital medium is here to stay. And teachers should be able to support students in learning both on site and remotely. Furthermore, it is important to implement opportunities offered by DS in everyday learning."

S: "It is important to try out new DS, evaluate whether the use of it has been successful, consider learning goals and then decide whether to use it further or not."

P: "Teachers in my school are not ready to teach digital competences to elementary school students in an integrated manner. They lack confidence in their ability to do so. We appointed a separate technology teacher for this purpose."

Nevertheless, we should not strive to force all educators to use DS. It is important to accept that the educator may have a negative attitude towards DS and allow them to conduct the lessons in a manner that the educator prefers, while emphasising that the students have the opportunity to learn the planned content in the learning format that is available to them. Furthermore, organising the learning process digitally, including creating new digital content for learning, is time-consuming and expensive, and there is not enough funding is available for it.

P: "Our school lost 2 good teachers, who were not ready to use DS in their lessons. Teachers should still have the opportunity to choose the methods they use for teaching. Even if those do not include digital solutions."

L: "If a teacher achieves good learning results using an analog format and is not ready to shift towards using DS it is important to provide them with the opportunity to keep working in their chosen way. We did not have a choice during the fully remote learning associated with the COVID-19 pandemic, but we do have it now."

S: "Teachers are very busy providing support for students to learn digitally, and creating digital solutions for it is very time-consuming. It is important to ask ourselves: will every teacher be ready to create digital content and will we be able to compensate them for it?"

DS in education promotes independence and autonomy, which is one of the reasons why students are likely to demand a different learning process after the experience of remote learning during the COVID-19 pandemic.

P: "Students learn a lot from screens, by using visual materials. We can fight against that and offer them books, but it is highly unlikely that it will work. We have to go further and develop pedagogically appropriate DS for learning."

P: "Teachers are intimidated by technology. Even though they might be willing to learn how to use DS, on an emotional level they are still unsure and feel a lot safer when teaching in an analog way."

There are no great differences in basic digital competences amongst educators of different age groups. Distinctive differences appear at higher levels of digital competences. Both younger and older educators can have a negative attitude towards using DS in education. It can be challenging for educators to understand the student learning experience, as it may, at times, profoundly differ from their own. This can have a negative impact on educator motivation to use DS.

P: "It is hard for some educators to empathise with students who prefer the digital medium for learning over learning from books, because they have not had such learning experiences themselves."

L: "There are no differences amongst educators in basic digital competence levels. Differences emerge at higher levels, such as programming, which not all students have and are unlikely to obtain soon. This competence is influenced to large extent by the attitude of students towards DS. Not all DS are easy to master; learning to use some of them takes effort and time, therefore persistence is of great importance."

As educators obtain positive experiences using technology, their desire to improve their digital competences grows. It is important to create these experiences consistently, gradually and at an appropriate pace, as well as offering the necessary support in the process of learning and applying DS, in order to gain the required sense of confidence amongst educators for using DS in their everyday practice. When teaching to improve digital competences, it is important to demonstrate them.

L: "If it is hard to use DS, then a negative attitude is formed towards those DS, which in turn can make the DS undesirable to use. If they are not used, competences to use them are not developed. If there are no competences in using DS—they will not be used. Therefore creating step by step experiences in using DS, along with developing a positive attitude towards using them, is crucial. There are DS that are intuitive and easy to master, but there are also those that can be cumbersome to learn, but they are still crucial for learning."

P: "It is crucial for teachers to feel safe when using DS for them to try them again. It is also crucial to be consistent in using DS in the agreed-upon ways and to implement them at an organisational level. What may seem very easy for digitally competent users, can be much more complicated for those developing their digital competences. This potentially causes more problems in further education for teachers when developing digital competences."

S: "It is important to demonstrate DS, and give time to try them out. Students and other teachers have to be taught how to learn new DS themselves and should be provided with the freedom to choose which DS to use for particular needs."

P: "Teachers are motivated to learn DS if they feel successful doing so. Small steps are crucial to provide this sense of success. Also teachers have to see the point of using DS to be invested in learning how to use them."

P: "Teachers are interested in professional development related to the development of their digital competences. They understand that this is and will be important, even though they are not always pleased with using DS in their practice."

There are also several risks involved when considering remote learning, including learning online. High school students were at a higher risk for dropping out of school during remote learning caused by COVID-19, some of them chose to work, rather than continue their education.

P: "We lost several high school students during the pandemic. They chose to work instead of continuing in education."

L: "There are students who have only benefited from remote learning. Meanwhile there are also those who have not learned some of the content. I am also concerned about the high school students who leave education in order to work. We lose the opportunity to support them."

There are elements of the learning process that are more difficult to implement successfully online, such as assessment. Some students lack the sufficient competences to organise their own learning and successfully take part in online learning. On the other hand, the

online learning process reveals better when the student is not involved in learning, which we notice less frequently in the classroom.

　　　P: "DS in online lessons, if structured accordingly, help to monitor the progress of every student. This is perhaps even more effective than in-class learning."

　　　S: "I can not assert that I have found a good way to evaluate students in the remote learning process. Motivation and self discipline are very important for students to be successful in the remote learning process. Not all students succeed. I lacked any information on the progress of some students."

## 6. Discussion

　　　The digitalization of education has been steadily increasing over recent years due to the wider variety of possibilities offered by new technologies. This trend only accelerated during the COVID-19 pandemic due to the demand for remote learning [42]. Students are mostly back in schools now, but the experiences gained by both teachers and their students during remote learning have shifted their perspective on learning [43]. Digital solutions have cemented their place as a viable and accessible format of learning not only in remote settings, but also as a supplement and integral part of face-to-face learning for both students and teachers [43,44]. This does not mean, however, that all parties accept these changes willingly. For some educators this new reality has proven to be too much to bear, and they have left the profession, as was revealed in our focus group discussion. For others it is a struggle to see digital solutions as part of their teaching arsenal and they would much rather stick to using methods better known and understood by them. Furthermore, educators who do choose to use digital solutions, tend to replicate existing learning experiences, rather than choosing digital solutions that are more characteristic of the virtual learning environment [28,29] which could result in a less qualitative and engaging learning process for students [30–32]. Stressing that all teachers should be allowed the freedom of choice when it comes to learning methods they pick for organising their lessons, experts admit that the digitalization of education is not something that can or should be stopped at this point. Instead, teachers should be offered continuous support for implementing digital solutions in their lessons meaningfully and be taught how to help their students master using different digital solutions as well. Modelling positive learning experiences for teachers using digital solutions is crucial to increase their understanding and motivation to use technologies in their lessons.

　　　Student self-assessment reveals that they rate their digital competences quite highly. However, experts admit that teachers from all age groups lack skills to use digital solutions in learning meaningfully and have negative attitudes towards using digital technologies in education, including those who are still studying or have just attained higher education and started work in school. Negative attitudes towards using technologies in learning are caused by multiple factors, including the lack of digital solutions suitable for teaching, the lack of technology-enhanced learning experiences for teachers themselves and the lack of qualitative professional development opportunities in this field. Another aspect is the fact that some students do not have sufficient self-regulation skills to successfully use digital technologies in learning. Additionally, there is a lack of funding for teachers to create digital learning content, which is often time-consuming. As attitudes are some of the leading factors in choosing to improve competences, these factors can also influence the actual level of teacher digital competences [45]. Of course, these differences of opinions could be attributed to the limitations of self-assessment instruments in determining competences. Alternatively, it could indicate that students may not realise the extent they need to comprehend digital solutions to conduct technology-enhanced learning effectively. Considering the speed at which technologies evolve it is also crucial to keep in mind that life long learning is important in this field. Teachers have to have access to professional development during their work in school to increase their digital competences constantly. This cannot be solved by higher education alone. Furthermore, digital competences should be seen as a school-level characteristic rather than that of individual teachers. Hence, schools, including

strategic leadership, need to become digitally competent in their approaches to structuring and organising resources and institutional infrastructures to facilitate their staff to do the same [12,24–27].

Digital competences are taking a key role in teacher education [6,7]. It is important to focus on pedagogical-digital competences in teacher education—including the competences of using and creating new digital solutions for education. Therefore, it is important that students are equipped with the digital competences necessary to implement digital technologies in the learning process. Master and doctoral students from educational sciences have self-evaluated their digital competences as high. Students of educational sciences have self-assessed their digital communications and cooperation sub-competences as the highest among all digital sub-competences, which can be linked to the importance of these sub-competences for remote work, commonly used as a learning form during the COVID pandemic. However, the most important digital sub-competence that is essential for the modern learning process is the creation of digital content, which students of educational sciences have evaluated as one of the least developed among all digital sub-competences. In the focus group discussion participants emphasised the importance of digital technologies in organising the learning process, specifically highlighting the role of creating digital content sub-competences to ensure a contemporary learning process. It was also pointed out that teachers are experiencing significant difficulties when using digital technologies, particularly struggling with digital content creation. This has been cited as one of the reasons why some teachers have even left their jobs. Therefore, in future research it would be necessary to assess not only the digital competences of masters and doctoral students from educational sciences, but also the digital competences of teachers, and to identify and provide necessary support to those teachers who lack sufficient digital competences.

The results of the study show that master and doctoral students of educational sciences have assessed their digital competences as relatively high, with mean values of self-assessment higher, compared to the digital competences of students from other fields of study, but that the difference is not considered to be statistically significant. This could imply that student learning in educational sciences enables students to develop their digital competences or that they lack the skills to evaluate themselves objectively. It should be taken in consideration that the accuracy of the self-assessment survey, which is related to the assessment form, is lower compared to objective or behavioural observations or ability tests because respondents' responses can be affected by their limited ability to remember specific examples of their behaviour, distorted memories of their past behaviour, and a general tendency to assess their competences higher than they actually are [34,35,46].

There is an ongoing concern about whether educators have sufficient skills to promote their student digital competences. In Latvia it is especially concerning, considering our Digital Economy and Society Index [47] is low in several key areas for digital competences in comparison to other countries. This research shows that students from the field of pedagogy assess their digital skills as high and their self-assessment is similar to those of students from other fields—there are no statistically significant differences. There is a reason to hope that these educators will be able to provide the necessary support for their students and possibly colleagues to strengthen the digital competence in both teaching and learning. It is important to support these students as well as organize widely accessible and meaningful professional development events for educators already working in this education field, because, as the focus group discussion participants stressed and research reveals [46–48], there is still much work to be done concerning the improvement of digital competences.

## 7. Conclusions

In conclusion, the COVID-19 pandemic and the experience of remote learning have led to a growing demand for a different approach to education, one that promotes independence and autonomy among students. While students from various study fields assess their digital competence similarly, indicating a generally high level of proficiency, there is still a need to emphasise the importance of pedagogical-digital competences in

educational programs, as highlighted by practitioners. Despite students' self-assessment, educators face challenges in implementing technology-enhanced learning due to a lack of digital competences, negative attitudes towards digitalization, and limited availability of support and resources. It is crucial to recognize that learning by doing and learning by trial and error are valuable formats for acquiring digital competences. To facilitate this, educational institutions must incorporate digital solutions into their daily practices. This approach, supported by previous research [8,9,24,49–51], proves effective in developing digital competences. By addressing these structural elements and providing necessary support, we can foster a positive attitude towards digitalization and ensure educators are equipped with the digital competences required for effective teaching in the digital age.

*Limitations and Recommendations for Further Research*

Limitations of this research include the use of a self-assessment tool in competence measurement. Other methods for research, such as observations, content analysis and digital competence tests could be applied to further explore student digital competence. The research scope was limited to graduate students, data on undergraduate student digital competence could allow for improved understanding of digital competence development during higher education studies.

**Author Contributions:** Conceptualization, E.S., G.L., A.O., L.D. and Z.R.; Methodology, E.S., G.L., L.D. and Z.R.; Validation, E.S., G.L., A.O. and Z.R.; Formal analysis, E.S., G.L. and L.D.; Investigation, Z.R.; Data curation, E.S. and G.L.; Writing—original draft, E.S., G.L. and A.O.; Writing—review & editing, E.S., G.L., A.O., L.D. and Z.R.; Visualization, E.S. and G.L.; Supervision, L.D. and Z.R.; Project administration, Z.R.; Funding acquisition, Z.R. All authors have read and agreed to the published version of the manuscript.

**Funding:** This research was funded by the project "Assessment of the Students' Competences in Higher Education and their Development Dynamics during Study Period" ESF 8.3.6.2. "Development of Education Quality Monitoring System" 8.3.6.2/17/I/001 (23-12.6/22/2).

**Institutional Review Board Statement:** The study was conducted in accordance with the Declaration of Helsinki, and approved by the Humanities and Social Sciences Research Ethics Committee of the University of Latvia (protocol code 71-46/35 and date of approval 8 of February 2023).

**Informed Consent Statement:** Informed consent was obtained from all subjects involved in the study.

**Data Availability Statement:** The data presented in this study can be available on request from the corresponding author in accordance with the interests of the University of Latvia.

**Conflicts of Interest:** The authors declare no conflict of interest.

**Appendix A**

**Table A1.** Rotated factor matrix (Kaiser-Varimax rotation) for each digital competences self-assessment statement.

| No | Statement | F1 | F2 | F3 | F4 | F5 | F6 | F7 | F8 |
|----|-----------|----|----|----|----|----|----|----|----|
| | Information literacy | | | | | | | | |
| 1. | Using keywords and filters when searching for information on the Internet | 0.708 | | | | | | | |
| 2. | When searching for information on the Internet, different search engines are used | | | | | | | 0.457 | |
| 3. | Using tags and specific formats when searching for information on the Internet | | | | | | | 0.641 | |
| 4. | Using advanced search strategies (e.g., search operators) to find reliable information on the Internet | | | | | | | 0.64 | |
| 5. | Using web feeds (e.g., RSS Feeds) to receive updated content of interest | | | | | | | 0.68 | |

**Table A1.** *Cont.*

| No | Statement | F1 | F2 | F3 | F4 | F5 | F6 | F7 | F8 |
|----|-----------|----|----|----|----|----|----|----|----|
| 6. | Awareness about new solutions for searching, storing and recovering information and ability to apply them to problems | | | | | | | 0.623 | |
| 7. | When evaluating information, recognition that not all information available on the Internet is reliable | 0.75 | | | | | | | |
| 8. | Comparing different sources to assess the reliability of the information found | 0.768 | | | | | | | |
| 9. | Assessing the validity and reliability of the information using criteria appropriate to the specific situation | 0.718 | | | | | | | |
| 10. | Identifying, organising, determining how and where to easily obtain data, information and content in a digital environment | 0.621 | | | | | | | |
| 11. | Saving, organising, storing files or content (such as text, pictures, music, video, web pages) and recovering stored or stored files or content in a structured environment | 0.6 | | | | | | | |
| 12. | Classifying information in a methodical way using folders to facilitate retrieval | 0.642 | | | | | | | |
| 13. | Making backup copies for stored information or files | | | | | | | | |
| 14. | Storing information found on the Internet in different formats | | | | | | | 0.455 | |
| 15. | Using information storage clouds and document management systems (e.g., Dropbox, Google Drive, SharePoint, OneDrive) | | 0.411 | | | | | | |
| | Communication and collaboration | | | | | | | | |
| 1. | Contacting others via IP Voice (e.g., Skype, WhatsApp, Facebook Messenger, Zoom, Google Hangouts), e-mail or chat using basic functions | 0.679 | | | | | | | |
| 2. | Transferring or sharing knowledge with others on the Internet | | 0.546 | | | | | | |
| 3. | Systematically using a wide range of communication tools (e-mail, chat, text messages, blogs, microbloggers, social networks), using different functions of these tools | 0.536 | 0.496 | | | | | | |
| 4. | Sharing files and content with simple sharing tools (such as OneDrive, Google Drive, Dropbox) | | 0.566 | | | | | | |
| 5. | Using collaboration tools and working with shared documents/files created by someone else | 0.471 | 0.58 | | | | | | |
| 6. | Using technologically sophisticated sharing functions (e.g., desktop sharing, application sharing, etc.) | | 0.585 | | | | | | |
| 7. | Using a wide range of online services (e.g., public services, internet banking, internet shopping) | 0.653 | | | | | | | |
| 8. | Developing and maintaining digital interaction tools and platforms (e.g., discussion platform, online collection of signatures, etc.) | | 0.594 | | | | | 0.448 | |
| 9. | Using online collaboration tools (social networks such as Facebook, LinkedIn, ResearchGate) for systematic collaboration | | 0.645 | | | | | | |
| 10. | In day-to-day systematic cooperation, using specialised cooperation tools and additional functions (e.g., electronic calendars, project management systems, etc.) | | 0.676 | | | | | | |
| 11. | Creating, managing content and organising collaboration using specialised collaboration tools | | 0.619 | | | | | | |
| 12. | On-line communication complies with the communication rules set out in a specific platform | 0.513 | | | | | | | |
| 13. | Adhering to generally accepted online communication rules in online communication | 0.52 | | | | | | | |

**Table A1.** *Cont.*

| No | Statement | F1 | F2 | F3 | F4 | F5 | F6 | F7 | F8 |
|----|-----------|----|----|----|----|----|----|----|----|
| 14. | Adhering to online communication rules, taking into account audience specificities (e.g., cultural and intergenerational differences) | 0.573 | | | | | | | |
| 15. | Using digital attributes and tools to create digital identity | | 0.509 | | | | | | |
| 16. | Taking action to protect reputation online | 0.469 | | | | | | | |
| 17. | Manipulating own data, created using digital tools, environment and technologies, by changing and refining them (e.g., deleting social network records, search engine indexed information about oneself) | | | | | | | | |
| Digital content creation | | | | | | | | | |
| 1. | Manging simple digital content with digital tools (e.g., text, tables, pictures, audio files) | 0.547 | | | | 0.488 | | | |
| 2. | Creating complex digital content in different formats (e.g., text, tables, pictures, audio files) | | | | | 0.538 | | | |
| 3. | Using tools/editors to create Web pages or memoirs using prepared templates (e.g., WordPress) | | | | | 0.479 | | | |
| 4. | Creating and/or transforming sophisticated media content in different formats and databases using different digital platforms, tools and environments | | | | | 0.535 | | | |
| 5. | Editing basic content that has already been created (for example, formatting text in a Word file) | 0.465 | | | | 0.569 | | | |
| 6. | Applying basic formatting (e.g., insert footnotes, charts, tables, etc.) to other or own content | 0.451 | | | | 0.604 | | | |
| 7. | Using additional formatting functions for different tools (such as e-mail merge, merging documents in different formats, using complex formulas, macros) | | | | | 0.475 | | | |
| 8. | Recognising that content may be subject to copyright when creating digital content (such as text, presentation, videos, etc.). | 0.404 | | | | 0.571 | | | |
| 9. | Referencing the primary sources of digital content and/or avoiding unauthorised use of copyrighted digital content | | | | | 0.59 | | | |
| 10. | Considering copyright restrictions and/or requirements when creating digital content | | | | | 0.522 | | | |
| 11. | Applying and transforming simple functions and settings for software and applications (e.g., changing the default settings) | | | | | 0.45 | | | |
| 12. | Using command line interface and/or ready application code to perform tasks | | | | | | 0.533 | | |
| 13. | Using programming languages (e.g., Python, Java, C #, C++, C, Swift, HTML, javascript, R, etc.) | | | | | | 0.512 | | |
| Safety | | | | | | | | | |
| 1. | Employing basic device protection measures with the assistance of IT specialists (e.g., antivirus and password usage). | | | | | | | | |
| 2. | Mitigating high-risk digital activities (e.g., downloading files from untrusted websites, opening suspicious email attachments, using strong passwords, etc.) | 0.478 | | | | | | | |
| 3. | Installing security programs for devices used to access the Internet (e.g., antivirus, firewall), regularly updating them | 0.417 | | 0.432 | | | | | |
| 4. | Using different passwords to access equipment, devices and digital services, periodically changing them, creating complex passwords | | | | | | | | |
| 5. | Frequently checking security configurations and systems for devices and/or applications that use, configure, or change firewall and security settings for their digital devices | | | 0.421 | | | 0.533 | | |

**Table A1.** *Cont.*

| No | Statement | F1 | F2 | F3 | F4 | F5 | F6 | F7 | F8 |
|---|---|---|---|---|---|---|---|---|---|
| 6. | Restoring the operation of the device without any other assistance if it has been infected with a virus | | | | | | 0.643 | | |
| 7. | Identifying unreliable information on the Internet | 0.456 | | | | | 0.499 | | |
| 8. | Taking precautionary measures to protect private information | 0.514 | | 0.423 | | | 0.405 | | |
| 9. | Systematically, critically building digital identity on the Internet and regularly tracking the digital footprint | | | 0.418 | | | 0.527 | | |
| 10. | Encrypting e-mails or files, using digital solutions to protect personal data and identity | | | 0.44 | | | 0.518 | | |
| 11. | Avoiding prolonged use of digital technologies, considering potential negative health effects | | | 0.686 | | | | | |
| 12. | Taking actions to mitigate the health risks associated with digital technology (e.g., blue light filters, ergonomic solutions, etc.) | | | 0.557 | | | | | |
| 13. | Identifying which websites or e-mail messages could be used for fraudulent purposes, applying filters to spam | | | 0.424 | | | | | |
| 14. | Balancing the use of digital technologies with other working and life activities | | | 0.679 | | | | | |
| 15. | Taking basic steps to save energy (for example, turns off computer) | | | 0.644 | | | | | |
| 16. | Taking systematic action to maximise the positive and minimise the negative impact of technologies on the environment (e.g., avoiding paper printing, using energy saving modes, choosing devices with less electricity consumption, etc.) | | | 0.677 | | | | | |
| 17. | Choosing digital technologies, taking into account the wider context of their environmental impact (ecological footprint in the production process, the reputation of the manufacturer in the field of environmental protection, sustainability of the materials used, etc.) | | | 0.646 | | | | | |
| | Problemsolving | | | | | | | | |
| 1. | Solving simple, casual technical problems (e.g., closing a program, restarting a computer, checking the internet connection) | 0.643 | | | 0.427 | | | | |
| 2. | Solving most common problems with digital technologies by exploring settings and program or tool options | 0.482 | | | 0.573 | | | | |
| 3. | Solving almost all of the challenges of using digital technologies and helping others | | | | 0.623 | 0.451 | | | |
| 4. | Choosing a digital tool according to specific needs and assessing its effectiveness | | | | 0.656 | | | | |
| 5. | Helping others choose the appropriate digital tool, device, application, software, or service to solve problems | | | | 0.67 | | | | |
| 6. | Finding support and help when having trouble using a new device or program | 0.452 | | | 0.542 | | | | |
| 7. | Using digital technologies to solve technical and non-technical problems | | | | 0.666 | | | | |
| 8. | Following technological developments and testing new tools | | | | 0.57 | | | | |
| 9. | Developing digital skills when external demand arises | 0.428 | | | 0.535 | | | | |
| 10. | Being aware of the incompleteness of their skills and working on skills development | 0.447 | | | 0.514 | | | | |
| 11. | Evaluating the potential future demand for digital competences, regularly and continually embracing new digital competences | | | | 0.546 | | | | |

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
