# Peer review of "Development of Education Field Student Digital Competences—Student and Stakeholders’ Perspective"

_sustainability, doi:10.3390/su15139895_

Round 1

Reviewer 1 Report

Dear editor

Dear Author

This study aim of this study is to analyze students' self-assessment of their digital competences, to compare the results of students studying education to those studying in other fields, and to identify the opinions of educational stakeholders regarding the most important digital competences, in order to develop recommendations for improving digital competence.

Before accepted this article for publication, here you can find my comment to improve the quality of manuscript.

Abstract section: there is lack of methodology and implications and contributions of this study. the abstract not interesting and monotonous. Please re-written the abstract.

 Literature review: literature review lack of references. author need to give a sub-title and explain in depth about the meaning of digital competencies, digital competencies in higher education, the role of digital competencies, the previous study about digital competencies, the important of digital competencies in detail.

Methodology section: why author not include undergraduate students in this study? according to T. T. Wijaya, Y. Cao, M. Bernard, I. F. Rahmadi, Z. Lavicza, and H. D. Surjono, “Factors influencing microgame adoption among secondary school mathematics teachers supported by structural equation modelling-based research,” Front. Psychol., vol. 13, no. September, pp. 1–16, 2022, doi: 10.3389/fpsyg.2022.952549, in perspective teacher, more important for undergraduate and master degree to have digital competencies. Their digital competencies have a connection with the quality of teaching and learning activities.

not clear. Why use 7 Likert scale?

when and how author collect the data? not clear.

How author analyze and sort the data? not clear.

How group discussion done? Not clear.

Line 194 can be concluded that the mean values of all digital sub-competences are relatively high and are higher than 5… this may too subjective, better if author added references.

This paper lacks limitations and recommendations section. Also, implications sections.

Conclusion section: conclusion should write with descriptive paragraph. Use the first conclusion paragraph for summarizing your main ideas, but try not to quote too much from the actual text of the descriptive article itself.

This manuscript less references. please added more references. The number of references proves author have been read a lot of literature and have a basic knowledge in this field.

Author Response

Dear reviewer!

We want to express our sincere appreciation for the time and effort you dedicated to reviewing our work. Your comments have been most helpful in improving our article! We will clarify the changes in further paragraphs.

Taking into consideration your concerns we shortened the explanatory part of the abstract. We added more detailed information about the methodology used in the research and elaborated on the implications and contributions of the research.

We added 14 more references to the literature review. As well as elaborated on the importance of digital competence for educators, including digital competence in higher education.

We agree that it would also be worth analyzing undergraduate digital competence. The boundaries of the study were in line with project goals and findings were used to raise awareness of the effectiveness of master's and doctoral studies for the development of digital competencies that are required to meet today's requirements. But we added this as a recommendation for further research.

In methodology, we explained the rationale behind choosing the 7-point Likert scale. The article was also supplemented by information on the time when the survey used for gathering data was available and how students could choose to fill it. Also, an additional step of the focus group discussion was added to the methodology to better depict the discussion format. We also revised the methodology section regarding data analysis. 

We also rewrote the conclusion section in a descriptive paragraph.

Please, see the changes added in the attached document.

Thank you again for your detailed review!

With warm regards, research team.

Reviewer 2 Report

This paper is the result of a research to analyse students' self-assessment of their digital competences. The focus is on students studying education in comparison to students of other fields, and on opinions of stakeholders.

The paper is clearly structured with a detailed presentation of the literature on this topic and with a clear description of the methodology and results.

I don't have further comments and suggestions for the publication of this paper. 

Author Response

Dear Reviewer!

We would like to express our sincere gratitude for your favorable response to our article titled "DEVELOPMENT OF DIGITAL COMPETENCES AMONG STUDENTS IN THE FIELD OF EDUCATION – PERSPECTIVES OF STUDENTS AND STAKEHOLDERS." Your positive feedback regarding the clear structure, comprehensive literature review, and detailed presentation of the methodology and results is greatly appreciated.

We are thankful for your assessment that no further comments or suggestions are necessary for the publication of this paper. Your validation signifies that the research was conducted effectively, and the findings were communicated clearly.

Your time and expertise in reviewing our work are highly valued. Your support and acknowledgment hold significant importance to us, and we are honored by your positive evaluation of the article.

Once again, we extend our heartfelt thanks for your valuable contribution to the publication process.

Sincerely,

Edīte Sarva , Gatis Lāma , Alise Oļesika , Linda Daniela , Zanda Rubene

Reviewer 3 Report

The article is well written, in an academic and scientific language and is an interesting topic for the scientific community. There are no apparent grammatical or spelling errors. The objectives are in clear agreement with the methodology. The conclusions follow directly from the development of the work, they attend to and are related to the purpose of the article and to the title.The citation of mainstream sources (books and magazine articles) predominates.

Author Response

Dear reviewer,

We would like to express our sincere gratitude for your positive evaluation of our article titled "DEVELOPMENT OF DIGITAL COMPETENCES AMONG STUDENTS IN THE FIELD OF EDUCATION – PERSPECTIVES OF STUDENTS AND STAKEHOLDERS." We appreciate your remarks highlighting the academic and scientific language used throughout the paper, as well as the article's relevance to the scientific community.

Your observation regarding the absence of grammatical or spelling errors is duly noted and reassuring. We strive to ensure the highest level of linguistic accuracy in our work.

We are pleased to hear that the objectives of our research align well with the chosen methodology. Your recognition of this agreement provides validation for our study design.

Furthermore, we appreciate your observation that the conclusions drawn directly stem from the development of our work, remaining consistent with the article's purpose and title. This serves as confirmation that our findings are in line with our initial intentions.

Your comment on the predominance of citations from mainstream sources, such as books and magazine articles, is duly acknowledged. We believe that drawing from well-established references strengthens the credibility and robustness of our research.

Once again, we sincerely thank you for your thoughtful evaluation and positive feedback. Your expert assessment and acknowledgment of the article's merits are greatly valued. 

With utmost appreciation,

Edīte Sarva , Gatis Lāma , Alise Oļesika , Linda Daniela , Zanda Rubene

Round 2

Reviewer 1 Report

the quality of the paper has been improved.

now paper ready to publish.

well done.